# StructBERT:
# Incorporating Language Structures into Pre-training for Deep Language Understanding

**Wei Wang, Bin Bi, Ming Yan, Chen Wu, Jiangnan Xia, Zuyi Bao, Liwei Peng and Luo Si**
Alibaba Group Inc.
`{hebian.ww, b.bi, ym119608, wuchen.wc, jiangnan.xjn, zuyi.bzy,`
`liwei.peng, luo.si}@alibaba-inc.com`

## Abstract

Recently, the pre-trained language model, BERT (and its robustly optimized version RoBERTa), has attracted a lot of attention in natural language understanding (NLU), and achieved state-of-the-art accuracy in various NLU tasks, such as sentiment classification, natural language inference, semantic textual similarity and question answering. Inspired by the linearization exploration work of Elman (Elman, 1990), we extend BERT to a new model, StructBERT, by incorporating language structures into pre-training. Specifically, we pre-train StructBERT with two auxiliary tasks to make the most of the sequential order of words and sentences, which leverage language structures at the word and sentence levels, respectively. As a result, the new model is adapted to different levels of language understanding required by downstream tasks.

The StructBERT with structural pre-training gives surprisingly good empirical results on a variety of downstream tasks, including pushing the state-of-the-art on the GLUE benchmark to 89.0 (outperforming all published models at the time of model submission), the F1 score on SQuAD v1.1 question answering to 93.0, the accuracy on SNLI to 91.7.

## 1 Introduction

A pre-trained language model (LM) is a key component in many natural language understanding (NLU) tasks such as semantic textual similarity (Cer et al., 2017), question answering (Rajpurkar et al., 2016) and sentiment classification (Socher et al., 2013). In order to obtain reliable language representations, neural language models are designed to define the joint probability function of sequences of words in text with self-supervised learning. Different from traditional word-specific embedding in which each token is assigned a global representation, recent work, such as Cove (McCann et al., 2017), ELMo (Peters et al., 2018), GPT (Radford et al., 2018) and BERT (Devlin et al., 2018), derives contextualized word vectors from a language model trained on a large text corpus. These models have been shown effective for many downstream NLU tasks.

Among the context-sensitive language models, BERT (and its robustly optimized version RoBERTa (Liu et al., 2019b)) has taken the NLP world by storm. It is designed to pre-train bidirectional representations by jointly conditioning on both left and right context in all layers and model the representations by predicting masked words only through the contexts. However, it does not make the most of underlying language structures.

According to Elman (Elman, 1990)'s study, the recurrent neural networks was shown to be sensitive to regularities in word order in simple sentences. Since language fluency is determined by the ordering of words and sentences, finding the best permutation of a set of words and sentences is an essential problem in many NLP tasks, such as machine translation and NLU (Hasler et al., 2017). Recently, word ordering was treated as LM-based linearization solely based on language models (Schmaltz et al., 2016). Schmaltz showed that recurrent neural network language models (Mikolov et al., 2010) with long short-term memory (Hochreiter & Schmidhuber, 1997) cells work effectively for word ordering even without any explicit syntactic information.

In this paper, we introduce a new type of contextual representation, StructBERT, which incorporates language structures into BERT pre-training by proposing two novel linearization strategies. Specifically, in addition to the existing masking strategy, StructBERT extends BERT by leveraging the structural information: word-level ordering and sentence-level ordering. We augment model pre-training with two new structural objectives on the inner-sentence and inter-sentence structures, respectively. In this way, the linguistic aspects (Elman, 1990) are explicitly captured during the pre-training procedure. With structural pre-training, StructBERT encodes dependency between words as well as sentences in the contextualized representation, which provides the model with better generalizability and adaptability.

StructBERT significantly advances the state-of-the-art results on a variety of NLU tasks, including the GLUE benchmark (Wang et al., 2018), the SNLI dataset (Bowman et al., 2015) and the SQuAD v1.1 question answering task (Rajpurkar et al., 2016). All of these experimental results clearly demonstrate StructBERT's exceptional effectiveness and generalization capability in language understanding.

We make the following major contributions:

- We propose novel structural pre-training that extends BERT by incorporating the word structural objective and the sentence structural objective to leverage language structures in contextualized representation. This enables the StructBERT to explicitly model language structures by forcing it to reconstruct the right order of words and sentences for correct prediction.

- StructBERT significantly outperforms all published state-of-the-art models on a wide range of NLU tasks at the time of model submission. This model extends the superiority of BERT, and boosts the performance in many language understanding applications such as semantic textual similarity, sentiment analysis, textual entailment, and question answering.

## 2 STRUCTBERT MODEL PRE-TRAINING

StructBERT builds upon the BERT architecture, which uses a multi-layer bidirectional Transformer network (Vaswani et al., 2017). Given a single text sentence or a pair of text sentences, BERT packs them in one token sequence and learns a contextualized vector representation for each token. Every input token is represented based on the word, the position, and the text segment it belongs to. Next, the input vectors are fed into a stack of multi-layer bidirectional Transformer blocks, which uses self-attention to compute the text representations by considering the entire input sequence.

The original BERT introduces two unsupervised prediction tasks to pre-train the model: i.e., a masked LM task and a next sentence prediction task. Different from original BERT, our StructBERT amplifies the ability of the masked LM task by shuffling certain number of tokens after word masking and predicting the right order. Moreover, to better understand the relationship between sentences, StructBERT randomly swaps the sentence order and predicts the next sentence and the previous sentence as a new sentence prediction task. In this way, the new model not only explicitly captures the fine-grained word structure in every sentence, but also properly models the inter-sentence structure in a bidirectional manner. Once the StructBERT language model is pre-trained with these two auxiliary tasks, we can fine-tune it on task-specific data for a wide range of downstream tasks.

### 2.1 INPUT REPRESENTATION

Every input $x$ is a sequence of word tokens, which can be either a single sentence or a pair of sentences packed together. The input representation follows that used in BERT (Devlin et al., 2018). For each input token $t_i$, its vector representation $\mathbf{x}_i$ is computed by summing the corresponding token embedding, positional embedding, and segment embedding. We always add a special classification embedding ([CLS]) as the first token of every sequence, and a special end-of-sequence ([SEP]) token to the end of each segment. Texts are tokenized to subword units by WordPiece (Wu et al., 2016) and absolute positional embeddings are learned with supported sequence lengths up to 512 tokens. In addition, the segment embeddings are used to differentiate a pair of sentences as in BERT.

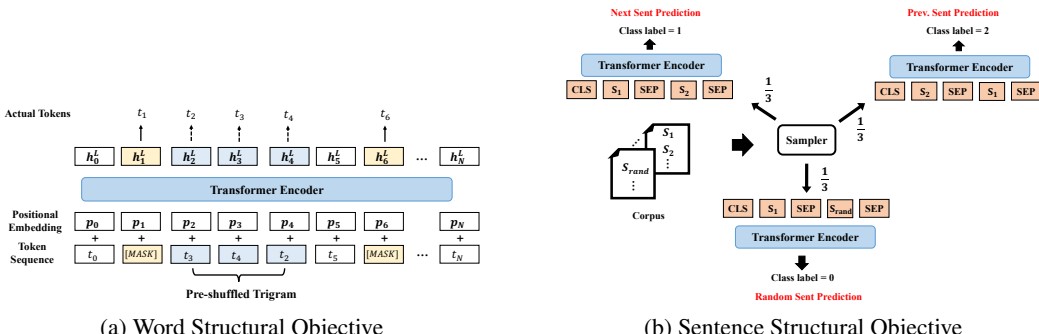

(a) Word Structural Objective          (b) Sentence Structural Objective

Figure 1: Illustrations of the two new pre-training objectives

## 2.2 TRANSFORMER ENCODER

We use a multi-layer bidirectional Transformer encoder (Vaswani et al., 2017) to encode contextual information for input representation. Given the input vectors $\mathbf{X} = \{\mathbf{x}_i\}_{i=1}^N$, an $L$-layer Transformer is used to encode the input as:

$$\mathbf{H}^l = Transformer_l(\mathbf{H}^{l-1}) \tag{1}$$

where $l \in [1, L]$, $\mathbf{H}^0 = \mathbf{X}$ and $\mathbf{H}^L = [\mathbf{h}_1^L, \cdots, \mathbf{h}_N^L]$. We use the hidden vector $\mathbf{h}_i^L$ as the contextualized representation of the input token $t_i$.

## 2.3 PRE-TRAINING OBJECTIVES

To make full use of the rich inner-sentence and inter-sentence structures in language, we extend the pre-training objectives of original BERT in two ways: ① word structural objective (mainly for the single-sentence task), and ② sentence structural objective (mainly for the sentence-pair task). We pre-train these two auxiliary objectives together with the original masked LM objective in a unified model to exploit inherent language structures.

### 2.3.1 WORD STRUCTURAL OBJECTIVE

Despite its success in various NLU tasks, original BERT is unable to explicitly model the sequential order and high-order dependency of words in natural language. Given a set of words in random order from a sentence, ideally a good language model should be able to recover this sentence by reconstructing the correct order of these words. To implement this idea in StructBERT, we supplement BERT's training objectives with a new word structural objective which endows the model with the ability to reconstruct the right order of certain number of intentionally shuffled word tokens. This new word objective is jointly trained together with the original masked LM objective from BERT.

Figure 1a illustrates the procedure of jointly training the new word objective and the masked LM objective. In every input sequence, we first mask 15% of all tokens at random, as done in BERT (Devlin et al., 2018). The corresponding output vectors $\mathbf{h}_i^L$ of the masked tokens computed by the bidirectional Transformer encoder are fed into a softmax classifier to predict the original tokens.

Next, the new word objective comes into play to take word order into consideration. Given the randomicity of token shuffling, the word objective is equivalent to maximizing the likelihood of placing every shuffled token in its correct position. More formally, this objective can be formulated as:

$$\arg\max_{\theta} \sum \log P(\text{pos}_1 = t_1, \text{pos}_2 = t_2, \ldots, \text{pos}_K = t_K | t_1, t_2, \ldots, t_K, \theta), \tag{2}$$

where $\theta$ represents the set of trainable parameters of StructBERT, and $K$ indicates the length of every shuffled subsequence. Technically, a larger $K$ would force the model to be able to reconstruct longer sequences while injecting more disturbed input. On the contrary, when $K$ is smaller, the model gets more undisturbed sequences while less capable of recovering long sequences. We decide to use trigrams (i.e., $K = 3$) for subsequence shuffling to balance language reconstructability and robustness of the model.

Specifically, as shown in Figure 1a, we randomly choose some percentage of trigrams from unmasked tokens, and shuffle the three words (e.g., $t_2$, $t_3$, and $t_4$ in the figure) within each of the trigrams. The output vectors of the shuffled tokens computed by the bidirectional Transformer encoder are then fed into a softmax classifier to predict the original tokens. The new word objective is jointly learned together with the masked LM objective in a unified pre-trained model with equal weights.

### 2.3.2 SENTENCE STRUCTURAL OBJECTIVE

The next sentence prediction task is considered easy for the original BERT model (the prediction accuracy of BERT can easily achieve 97%-98% in this task (Devlin et al., 2018)). We, therefore, extend the sentence prediction task by predicting both the next sentence and the previous sentence, to make the pre-trained language model aware of the sequential order of the sentences in a bidirectional manner.

As illustrated in Figure 1b, given a pair of sentences $(S_1, S_2)$ as input, we predict whether $S_2$ is the next sentence that follows $S_1$, or the previous sentence that precedes $S_1$, or a random sentence from a different document. Specifically, for the sentence $S_1$, $\frac{1}{3}$ of the time we choose the text span that follows $S_1$ as the second sentence $S_2$, $\frac{1}{3}$ of the time the previous sentence ahead of $S_1$ is selected, and $\frac{1}{3}$ of the time a sentence randomly sampled from the other documents is used as $S_2$. The two sentences are concatenated together into an input sequence with the separator token [SEP] in between, as done in BERT. We pool the model output by taking the hidden state corresponding to the first token [CLS], and feed the encoding vector of [CLS] into a softmax classifier to make a three-class prediction.

### 2.4 PRE-TRAINING SETUP

The training objective function is a linear combination of the word structural objective and the sentence structural objective. For the masked LM objective, we followed the same masking rate and settings as in BERT (Devlin et al., 2018). 5% of trigrams are selected for random shuffling.

We used documents from English Wikipedia (2,500M words) and BookCorpus (Zhu et al., 2015) as pre-training data, following the preprocessing and the WordPiece tokenization from (Devlin et al., 2018). The maximum length of input sequence was set to 512.

We ran Adam with learning rate of 1e-4, $\beta_1 = 0.9$, $\beta_2 = 0.999$, L2 weight decay of 0.01, learning rate warm-up over the first 10% of the total steps, and linear decay of the learning rate. We set a dropout probability of 0.1 for every layer. The gelu activation (Hendrycks & Gimpel, 2016) was used as done in GPT (Radford et al., 2018).

We denote the number of Transformer block layers as $L$, the size of hidden vectors as $H$, and the number of self-attention heads as $A$. Following the practice of BERT, We primarily report experimental results on the two model sizes:

**StructBERTBase**: $L = 12$, $H = 768$, $A = 12$, Number of parameters= 110M

**StructBERTLarge**: $L = 24$, $H = 1024$, $A = 16$, Number of parameters= 340M

Pre-training of StructBERT was performed on a distributed computing cluster consisting of 64 Telsa V100 GPU cards. For the StructBERTBase, we ran the pre-training procedure for 40 epochs, which took about 38 hours, and the training of StructBERTLarge took about 7 days to complete.

## 3 EXPERIMENTS

In this section, we report results of StructBERT on a variety of downstream tasks including General Language Understanding Evaluation (GLUE benchmark), Standford Natural Language inference (SNLI corpus) and extractive question answering (SQuAD v1.1).

Following BERT's practice, during fine-tuning on downstream tasks, we performed a grid search or an exhaustive search (depending on the data size) on the following sets of parameters and chose the model that performed the best on the dev set. All the other parameters remain the same as those in pre-training:

*Batch size*: 16, 24, 32; *Learning rate*: 2e-5, 3e-5, 5e-5; *Number of epochs*: 2, 3; *Dropout rate*: 0.05, 0.1

| System | CoLA | SST-2 | MRPC | STS-B | QQP | MNLI | QNLI | RTE | WNLI | AX | Avg. |
|---|---|---|---|---|---|---|---|---|---|---|---|
| | 8.5k | 67k | 3.5k | 5.7k | 363k | 392k | 108k | 2.5k | 634 | | |
| Human Baseline | 66.4 | 97.8 | 86.3/80.8 | 92.7/92.6 | 59.5/80.4 | 92.0/92.8 | 91.2 | 93.6 | 95.9 | - | |
| BERTLarge [1] | 60.5 | 94.9 | 89.3/85.4 | 87.6/86.5 | 72.1/89.3 | 86.7/85.9 | 92.7 | 70.1 | 65.1 | 39.6 | 80.5 |
| BERT on STILTs [2] | 62.1 | 94.3 | 90.2/86.6 | 88.7/88.3 | 71.9/89.4 | 86.4/85.6 | 92.7 | 80.1 | 65.1 | 28.3 | 82.0 |
| SpanBERT [3] | 64.3 | 94.8 | 90.9/87.9 | 89.9/89.1 | 71.9/89.5 | 88.1/87.7 | 94.3 | 79.0 | 65.1 | 45.1 | 82.8 |
| Snorkel MeTaL [4] | 63.8 | 96.2 | 91.5/88.5 | 90.1/89.7 | 73.1/89.9 | 87.6/87.2 | 93.9 | 80.9 | 65.1 | 39.9 | 83.2 |
| MT-DNN++ [5] | 65.4 | 95.6 | 91.1/88.2 | 89.6/89.0 | 72.7/89.6 | 87.9/87.4 | 95.8 | 85.1 | 65.1 | 41.9 | 83.8 |
| MT-DNN* [5] | 65.4 | 96.5 | 92.2/89.5 | 89.6/89.0 | 73.7/89.9 | 87.9/87.4 | 96.0 | 85.7 | 65.1 | 42.8 | 84.2 |
| StructBERTBase | 57.2 | 94.7 | 89.9/86.1 | 89.8/89.6 | 72.0/89.6 | 85.5/84.6 | 92.6 | 76.9 | 65.1 | 39.0 | 80.9 |
| StructBERTLarge | 65.3 | 95.2 | 92.0/89.3 | 90.3/89.4 | 74.1/90.5 | 88.0/87.7 | 95.7 | 83.1 | 65.1 | 43.6 | 83.9 |
| StructBERTLarge* | 68.6 | 95.2 | 92.5/90.1 | 91.1/90.6 | 74.4/90.7 | 88.2/87.9 | 95.7 | 83.1 | 65.1 | 43.9 | 84.5 |
| XLNet* [6] | 67.8 | 96.8 | 93.0/90.7 | 91.6/91.1 | 74.2/90.3 | 90.2/89.8 | 98.6 | 86.3 | 90.4 | 47.5 | 88.4 |
| RoBERTa* [7] | 67.8 | 96.7 | 92.3/89.8 | 92.2/91.9 | 74.3/90.2 | 90.8/90.2 | 98.9 | 88.2 | 89.0 | 48.7 | 88.5 |
| Adv-RoBERTa* | 68.0 | 96.8 | 93.1/90.8 | 92.4/92.2 | **74.8/90.3** | **91.1/90.7** | 98.8 | **88.7** | 89.0 | **50.1** | 88.8 |
| StructBERTRoBERTa* | **69.2** | **97.1** | **93.6/91.5** | **92.8/92.4** | 74.4/90.7 | 90.7/90.3 | **99.2** | 87.3 | **89.7** | 47.8 | **89.0** |

Table 1: Results of published models on the GLUE test set, which are scored by the GLUE evaluation server. The number below each task denotes the number of training examples. The state-of-the-art results are in bold. All the results are obtained from `https://gluebenchmark.com/leaderboard` (StructBERT submitted under a different model name ALICE). * indicates the ensemble model. Model references: [1]: (Devlin et al., 2018); [2]: (Phang et al., 2018); [3]: (Joshi et al., 2019); [4]: (Ratner et al., 2017); [5]: (Liu et al., 2019a); [6]: (Yang et al., 2019b); [7]: (Liu et al., 2019b).

## 3.1 GENERAL LANGUAGE UNDERSTANDING

### 3.1.1 GLUE BENCHMARK

The General Language Understanding Evaluation (GLUE) benchmark (Wang et al., 2018) is a collection of nine NLU tasks, covering textual entailment (RTE (Bentivogli et al., 2009) and MNLI (Williams et al., 2017)), question-answer entailment (QNLI (Wang et al., 2018)), paraphrase (MRPC (Dolan & Brockett, 2005)), question paraphrase (QQP), textual similarity (STS-B (Cer et al., 2017)), sentiment (SST-2 (Socher et al., 2013)), linguistic acceptability (CoLA), and Winograd Schema (WNLI (Levesque et al., 2012)).

On the GLUE benchmark, given the similarity of MRPC/RTE/STS-B to MNLI, we fine-tuned StructBERT on MNLI before training on MRPC/RTE/STS-B data for the respective tasks. This follows the two-stage transfer learning STILTs introduced in (Phang et al., 2018). For all the other tasks (i.e., RTE, QNLI, QQP, SST-2, CoLA and MNLI), we fine-tuned StructBERT for each single task only on its in-domain data.

Table 1 presents the results of published models on the GLUE test set obtained from the official benchmark evaluation server. Our StructBERTLarge ensemble suppressed all published models (excluding RoBERTa ensemble and XLNet ensemble) on the average score, and performed the best among these models in six of the nine tasks. In the most popular MNLI task, our StructBERTLarge single model improved the best result by 0.3%/0.5%, since we fine-tuned MNLI only on its in-domain data, this improvement is entirely attributed to our new training objectives. The most significant improvement over BERT was observed on CoLA (4.8%), which may be due to the strong correlation between the word order task and the grammatical error correction task. In the SST-2 task, our model improved over BERT while performed worse than MT-DNN did, which indicates that sentiment analysis based on single sentences benefits less from the word structural objective and sentence structural objective.

| Model | GPT | BERT | MT-DNN | SJRC | StructBERTLarge |
|---|---|---|---|---|---|
| Dev | - | 90.1 | 91.4 | - | **92.2** |
| Test | 89.9 | 90.8 | 91.1 | 91.3 | **91.7** |

Table 2: Accuracy (%) on the SNLI dataset.

| System | Dev set | | Test set | |
|---|---|---|---|---|
| | EM | F1 | EM | F1 |
| Human | - | - | 82.3 | 91.2 |
| XLNet(single+DA)  (Yang et al., 2019b) | 88.9 | 94.5 | 89.9 | 95.0 |
| BERT(ensemble+DA)  (Devlin et al., 2018) | 86.2 | 92.2 | 87.4 | 93.2 |
| KT-NET(single)  (Yang et al., 2019a) | 85.1 | 91.7 | 85.9 | 92.4 |
| BERT(single+DA)  (Devlin et al., 2018) | 84.2 | 91.1 | 85.1 | 91.8 |
| QANet(ensemble+DA) (Yu et al., 2018) | - | - | 84.5 | 90.5 |
| StructBERTLarge (single) | 85.2 | 92.0 | - | - |
| StructBERTLarge (ensemble) | 87.0 | 93.0 | - | - |

Table 3: SQuAD results. The StructBERTLarge ensemble is 10x systems which use different pre-training checkpoints and fine-tuning seeds.

With pre-training on large corpus, XLNet ensemble and RoBERTa ensemble outperformed all published models including our StructBERTLarge ensemble. To take advantage of the large data which RoBERTa is trained on, we continued pre-training with our two new objectives from the released RoBERTa model, named StructBERTRoBERTa. At the time of model submission, our StructBERTRoBERTa ensemble, which was submitted under a different name ALICE, achieved the best performance among all published models including RoBERTa and XLNet on the leaderboard, creating a new state-of-the-art result of 89.0% on the average GLUE score. It demonstrates that the proposed objectives are able to improve language models in addition to BERT.

### 3.1.2   SNLI

Natural Language Inference (NLI) is one of the important tasks in natural language understanding. The goal of this task is to test the ability of the model to reason the semantic relationship between two sentences. In order to perform well on an NLI task, a model needs to capture the semantics of sentences, and thus to infer the relationship between a pair of sentences: entailment, contradiction or neutral.

We evaluated our model on the most widely used NLI dataset: The Stanford Natural Language Inference (SNLI) Corpus (Bowman et al., 2015), which consists of 549,367/9,842/9,824 premise-hypothesis pairs in train/dev/test sets and target labels indicating their relations. We performed a grid search on the sets of parameters, and chose the model that performed best on the dev set.

Table 2 shows the results on the SNLI dataset of our model with other published models. StructBERT outperformed all existing systems on SNLI, creating new state-of-the-art results 91.7%, which amounts to 0.4% absolute improvement over the previous state-of-the-art model SJRC and 0.9% absolute improvement over BERT. Since the network architecture of our model is identical to that of BERT, this improvement is entirely attributed to the new pre-training objectives, which justifies the effectiveness of the proposed tasks of word prediction and sentence prediction.

### 3.2   EXTRACTIVE QUESTION ANSWERING

SQuAD v1.1 is a popular machine reading comprehension dataset consisting of 100,000+ questions created by crowd workers on 536 Wikipedia articles (Rajpurkar et al., 2016). The goal of the task is to extract the right answer span from the corresponding paragraph given a question.

We fine-tuned our StructBERT language model on the SQuAD dataset for 3 epochs, and compared the result against the state-of-the-art methods on the official leaderboard [1], as shown in Table 3. We can see that even without any additional data augmentation (DA) techniques, the proposed StructBERT model was superior to all published models except XLNet+DA on the dev set. [2]. With data augmentation and large corpus used during pre-training, XLNet+DA outperformed our StructBERT which did not

---

[1] https://rajpurkar.github.io/SQuAD-explorer/

[2] We have submitted the model under the name of ALICE to the SQuAD v1.1 CodaLab for evaluation on the test set. However, due to crash of the Codalab evaluation server, we have not got our test result back yet at the time of paper submission. We will update the result once it is announced.

| Task | CoLA (Acc) | SST-2 (Acc) | MNLI (Acc) | SNLI (Acc) | QQP (Acc) | SQuAD (F1) |
|---|---|---|---|---|---|---|
| StructBERTBase | **85.8** | **92.9** | **85.4** | 91.5 | **91.1** | **90.6** |
| -word structure | 81.7 | 92.7 | 85.2 | **91.6** | 90.7 | 90.3 |
| -sentence structure | 84.9 | **92.9** | 84.1 | 91.1 | 90.5 | 89.1 |
| BERTBase | 80.9 | 92.7 | 84.1 | 91.3 | 90.4 | 88.5 |

Table 4: Ablation over the pre-training objectives using StructBERTBase architecture. Every result is the average score of 8 runs with different random seeds (the MNLI accuracy is the average score of the matched and mis-matched settings).

use data augmentation or large pre-training corpus. It demonstrates the effectiveness of the proposed pre-trained StructBERT in modeling the question-paragraph relationship for extractive question answering. Incorporating the word and sentence structures significantly improves the understanding ability in this fine-grained answer extraction task.

### 3.3 EFFECT OF DIFFERENT STRUCTURAL OBJECTIVES

We have demonstrated the strong empirical results of the proposed model on a variety of downstream tasks. In the StructBERT pre-training, the two new structural prediction tasks are the most important components. Therefore, we conducted an ablation study by removing one structural objective from pre-training at a time to examine how the two structural objectives influence the performance on various downstream tasks.

Results are presented in Table 4. From the table, we can see that: (1) the two structural objectives were both critical to most of the downstream tasks, except for the word structural objective in the SNLI task. Removing any word or sentence objective from pre-training always led to degraded performance in the downstream tasks. The StructBERT model with structural pre-training consistently outperformed the original BERT model, which shows the effectiveness of the proposed structural objectives. (2) For the sentence-pair tasks such as MNLI, SNLI, QQP and SQuAD, incorporating the sentence structural objective significantly improved the performance. It demonstrates the effect of inter-sentence structures learned by pre-training in understanding the relationship between sentences for downstream tasks. (3) For the single-sentence tasks such as CoLA and SST-2, the word structural objective played the most important role. Especially in the CoLA task, which is related to the grammatical error correction, the improvement was over 5%. The ability of reconstructing the order of words in pre-training helped the model better judge the acceptability of a single sentence.

We also studied the effect of both structural objectives during self-supervised pre-training. Figure 2 illustrates the loss and accuracy of word and sentence prediction over the number of pre-training steps for StructBERTBase and BERTBase. From the two sub-figures on top, it is observed that

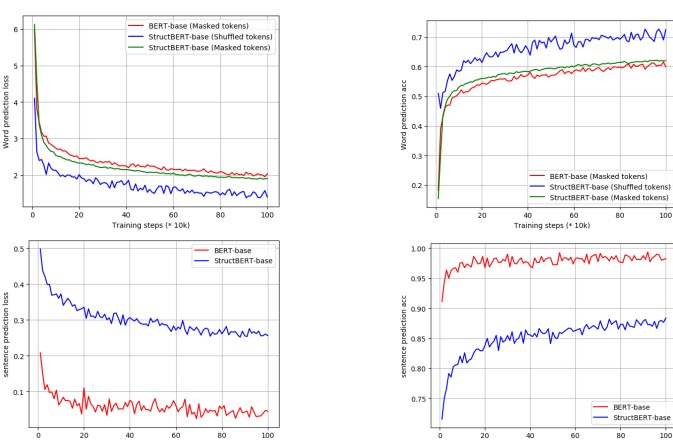

Figure 2: Loss and accuracy of word and sentence prediction over the number of pre-training steps

compared with BERT, the augmented shuffled token prediction in StructBERT's word structural objective had little effect on the loss and accuracy of masked token prediction. On the other hand, the integration of the simpler task of shuffled token prediction (lower loss and higher accuracy) provides StructBERT with the capability of word reordering. In contrast, the new sentence structural objective in StructBERT leads to a more challenging prediction task than that in BERT, as shown in the two figures at the bottom. This new pre-training objective enables StructBERT to exploit inter-sentence structures, which benefits sentence-pair downstream tasks.

## 4 RELATED WORK

### 4.1 CONTEXTUALIZED LANGUAGE REPRESENTATION

A word can have different semantics depending on the its context. Contextualized word representation is considered to be an important part of modern NLP research, with various pre-trained language models (McCann et al., 2017; Peters et al., 2018; Radford et al., 2018; Devlin et al., 2018) emerging recently. ELMo (Peters et al., 2018) learns two unidirectional LMs based on long short-term memory networks (LSTMs). A forward LM reads the text from left to right, and a backward LM encodes the text from right to left. Following the similar idea of ELMo, OpenAI GPT (Radford et al., 2018) expands the unsupervised language model to a much larger scale by training on a giant collection of free text corpora. Different from ELMo, it builds upon a multi-layer Transformer (Vaswani et al., 2017) decoder, and uses a left-to-right Transformer to predict a text sequence word-by-word.

In contrast, BERT (Devlin et al., 2018) (as well as its robustly optimized version RoBERTa (Liu et al., 2019b)) employs a bidirectional Transformer encoder to fuse both the left and the right context, and introduces two novel pre-training tasks for better language understanding. We base our LM on the architecture of BERT, and further extend it by introducing word and sentence structures into pre-training tasks for deep language understanding.

### 4.2 WORD & SENTENCE ORDERING

The task of linearization aims to recover the original order of a shuffled sentence (Schmaltz et al., 2016). Part of larger discussion as to whether LSTMs are capturing syntactic phenomena linearization, is standardized in a recent line of research as a method useful for isolating the performance of text-to-text generation (Zhang & Clark, 2015) models. Recently, Transformers have emerged as a powerful architecture for learning the latent structure of language. For example, Bidirectional Transformers (BERT) has reduced the perplexity for language modeling task. We revisit Elman's question by applying BERT to the word-ordering task, without any explicit syntactic approaches, and find that pre-trained language models are effective for various downstream tasks with linearization.

Many important downstream tasks such as STS and NLI (Wang et al., 2018) are based on understanding the relationship between two text sentences, which is not directly captured by language modeling. While BERT (Devlin et al., 2018) pre-trains a binarized next sentence prediction task to understand sentence relationships, we take one step further and treat it as a sentence ordering task. The goal of sentence ordering is to arrange a set of sentences into a coherent text in a clear and consistent manner, which can be viewed as a ranking problem (Chen et al., 2016). The task is general and yet challenging, and once is especially important for natural language generation (Reiter & Dale, 1997). Text should be organized according to the following properties: rhetorical coherence, topical relevancy, chronological sequence, and cause-effect. In this work, we focus on what is arguably the most basic characteristics of a sequence: their order. Most of prior work on sentence ordering was part of the study of downstream tasks, such as multi-document summarization (Bollegala et al., 2010). We revisit this problem in the context of language modeling as a new sentence prediction task.

## 5 CONCLUSION

In this paper, we propose novel structural pre-training which incorporates word and sentence structures into BERT pre-training. A word structural objective and a sentence structural objective are introduced as two new pre-training tasks for deep understanding of natural language in different granularities. Experimental results demonstrate that the new StructBERT model can obtain new state-of-the-art

results in a variety of downstream tasks, including the popular GLUE benchmark, the SNLI Corpus and the SQuAD v1.1 question answering.

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
