# OpenReview forum: "StructBERT: Incorporating Language Structures into Pre-training for Deep Language Understanding"
_ICLR.cc/2020/Conference — Accept (Poster)_

### Official Review · AnonReviewer1 · 2019-10-23
**Official Blind Review #1**

**Rating:** 3

**Review:**

This paper proposes to use additional structures within and between sentences for pre-training BERT. The basic idea is to shuffle either some n-grams within sentences or the sentences in texts, then train the model to predict the correct orders. Experiments in this work show that, with this additional training objective, the proposed pre-trained model, StructBERT, obtains good performance on the tasks including natural language understanding and question answering.

Overall, I think the experiments and results in this work are not sufficient enough to support the claim:

- It is necessary to show the performance of BERT only trained with the proposed word and sentence objectives. Otherwise, it is not clear how much benefit the model can get from them and the work is basically incremental.
- Some justification is needed about why choosing trigrams and why 5% is a good number of sampling trigrams from texts

Besides, there are some recent work on analyzing why BERT encodes any linguistic properties of texts, for example

- Goldberg. Assessing BERT's syntactic abilities. 2019
- Tenny et al. BERT Rediscovers the Classical NLP Pipeline. ACL 2019
- Tenny et al. What do you learn from context? ICLR 2019

All of them show positive results on BERT can capture some syntactic information from text automatically. Which makes me wonder why the simple additional training objective proposed in this work can still lead to performance improvement. Is there an explanation?


**Experience Assessment:**

I have published in this field for several years.

**Review Assessment: Checking Correctness Of Derivations And Theory:**

I carefully checked the derivations and theory.

**Review Assessment: Checking Correctness Of Experiments:**

I carefully checked the experiments.

**Review Assessment: Thoroughness In Paper Reading:**

I read the paper thoroughly.

---

> ### Author Response · Authors · 2019-11-15
> **Response to Reviewer #1**
>
> Thank you so much for going through the paper carefully and providing positive and useful feedback to our work!
> Please see our responses below:
>
> C1. Thanks for the suggestion. In StructBERT, the new sentence objective is designed to replace the Next Sentence Prediction (NSP) objective in original BERT, while the new word objective is a supplement to the masked LM objective. We will pretrain only with the new sentence and word objectives (without masked LM) to study how well it performs compared with BERT.
>
> C2. Following BERT's methodology, we experimented with different configurations of shuffled N-grams  and sampling rates, and found out the best setting of trigrams and 5% sampling rate. Analysis of the experiments will be detailed in the final version to justify our configuration choices.
>
> C3. Although original BERT can capture some syntactic information from text, we believe that our new structural objectives can inject more capability into the language model: 1) The new word objective forces the model to correct locally shuffled trigrams, and thus enhances its capability in modeling local syntax. Besides, the new word objective used for word ordering can be effective in controlling local fluency, which is also indicated in [1].  2) The new sentence objective, on the other hand, enables the model to capture discourse-level coherence properties between sentences (e.g., strengthening, contrast and causality). Similar findings are also reported in [2, 3].
>
> [1] Discriminative Syntax-Based Word Ordering for Text Generation, Zhang and Clark 2015
> [2] Discourse-Based Objectives for Fast Unsupervised Sentence Representation Learning, Jernite et al. 2017
> [3] ALBERT: A Lite BERT for Self-supervised Learning of Language Representations, Lan et al. 2019

---

### Official Review · AnonReviewer2 · 2019-10-24
**Official Blind Review #2**

**Rating:** 8

**Review:**

This paper proposed a new pre-trained language model based on BERT, called StructBERT. The key contributions are the two new pre-train objectives, (1) word structural objective, where the goal is to reconstruct the right order of intentionally shuffled word tokens, and (2) sentence structural objective, a three-class sentence-pair prediction, either the 2nd sentence precedes the 1st, the 2nd succeeds the 1st, or the 2nd is randomly selected. Unlike the original NSP (next sentence prediction) task, which is simple but tends out to be not so helpful in many downstream tasks, both proposed pre-train objectives seem to be rather useful in benchmarks tested in the paper, including GLUE, SNLI, and SQuAD.

The paper is well written and understandable for anyone who has a basic background about BERT or pre-train. The experimental results are impressive. Some of my questions / suggestions:

- The two auxiliary tasks are evidently helpful. I wonder what intuition/theory leads to the selection of these two tasks? If the authors have test multiple other tasks that were not as helpful, it is also interesting to know them.

- The wording of the text should be revised to reflect the up-to-date leaderboard results. Personally, I don't think the leaderboard results are that critical, but just want to make sure the writing is accurate at the time of publishing.

- Please also update the results from SQuAD 1.1 CodaLab.

**Experience Assessment:**

I have published one or two papers in this area.

**Review Assessment: Checking Correctness Of Derivations And Theory:**

N/A

**Review Assessment: Checking Correctness Of Experiments:**

I assessed the sensibility of the experiments.

**Review Assessment: Thoroughness In Paper Reading:**

I read the paper thoroughly.

---

> ### Author Response · Authors · 2019-11-15
> **Response to Reviewer #2**
>
> Thank you so much for going through the paper carefully and providing such a positive feedback. Please see below our response to your comments:
>
> C1. The intuition of word ordering task is from the task of Grammatical error correction, while the intuition of sentence ordering task is inspired from the discourse-level coherence property and causal relationship between the natural  sentences. Yes, we have also tested some other tasks: 1) mask only entities or nouns, 2) increase the mask rate, 3) predict the next sentence, nonadjacent sentence in the same document, and random sentence from another document. But we did not observe more improvement.
>
> C2. We agree with the reviewer about this and it has been fixed accordingly.
>
> C3. We have been in touch with SQuAD's administrator to evaluate our submitted model and update its score on the leaderboard. This process involves much manual effort from both us and the administrator. We have not got the updated score from the administrator yet. We will update our results on SQuAD upon receipt.

---

### Official Review · AnonReviewer3 · 2019-10-25
**Official Blind Review #3**

**Rating:** 6

**Review:**

This paper introduces two new tasks for large scale language model pretraining: trigram word unscrambling and contextual sentence ordering. Using these tasks to pretrain on top of masked language modelling shows improvements when the resulting model is finetuned on downstream tasks. The proposed tasks are simple to implement, and particularly the sentence ordering task is an improvement over the original BERT next sentence task, which is widely regarded as too simple to drive learning good representations. For this reason, I recommend acceptance of this paper.

Some minor quibbles:
1) Structure in language usually means syntactic structure. How does unscrambling word trigrams help uncover syntactic structure? The references to Elman 1990 also don't serve to clarify anything, I suggest that they are removed.
2) Some prior work on word ordering (e.g. [1] and older papers cited therein) is missing.
3) The permutation objective seems very similar to the XLNet objective. Could the authors elaborate more on this in the paper?
4) Did the authors try with other n-gram shuffling orders?
5) The sentence ordering task has been used previously (e.g. [2]).
6) Table 1 overhangs the right margin.

References:
[1] Discriminative Syntax-Based Word Ordering for Text Generation, Zhang and Clark 2015
[2] Discourse-Based Objectives for Fast Unsupervised Sentence Representation Learning, Jernite et al. 2017

**Experience Assessment:**

I have read many papers in this area.

**Review Assessment: Checking Correctness Of Derivations And Theory:**

N/A

**Review Assessment: Checking Correctness Of Experiments:**

I carefully checked the experiments.

**Review Assessment: Thoroughness In Paper Reading:**

I read the paper at least twice and used my best judgement in assessing the paper.

---

> ### Author Response · Authors · 2019-11-15
> **Response to Reviewer #3**
>
> Thank you so much for going through the paper carefully and providing positive and useful feedback to our work!
> Please see our responses below:
>
> C1. Thanks for your valuable suggestion. In our paper, structure refers to the word and sentence ordering inherent in natural language. Word trigrams are unscrambled to uncover the word ordering structure. We understand that the references to Elman 1990 are not very effective in clarification, and will remove them as suggested.
>
> C2. Thanks for the comment. We will add the missing citations accordingly.
>
> C3. The pretraining objective in XLNet belongs to the autoregressive (AR) language modeling, where they use all permutations of the factorization order to approximate the AR objective. By contrast, our new word objective is still an autoencoding (AE) one, which is designed to model local language structures by incorporating word  ordering into pretraining. Moreover, our word objective also differs from XLNet's in that ours permutates word order while XLNet's objective permutates factorization order. Specifically, the order of words in XLNet does not change given their fixed positional embeddings. In contrast, StructBERT shuffles words by changing their positions in text. We will include the elaboration in our final version.
>
> C4. We did try bigram and four-gram shuffling orders, but did not observe further improvement over trigram shuffling. We speculate that shuffling less words (bigrams) cannot take full advantage of the word ordering structure, while shuffling more words (4+-grams) can introduce more noise and harm the robustness of the model.
>
> C5. The Binary Ordering of Sentences in [2]  models the ordering of two consecutive sentences. Despite the similarity, our new objective differs in two ways: 1) It is defined on textual segments rather than natural sentences. 2) It is 3-way classification of segments while the objective in [2] determines binary ordering of sentences. We will add this work and its difference from ours in our final version.
>
> C6. It has been fixed.

---

### Decision · Program_Chairs · 2019-12-19

**Decision:**

Accept (Poster)

**Comment:**

This paper proposes a pair of complementary word- and sentence-level pretraining objectives for BERT-style models, and shows that they are empirically effective, especially when used with an already-pretrained RoBERTa model.

Work of this kind has been extremely impactful in NLP, and so I'm somewhat biased toward acceptance: If this isn't published, it seems likely that other groups will go to the trouble to replicate roughly these experiments. However, I think the paper is borderline. Reviewers were impressed by the results, but not convinced that the ablations and analyses were sufficient to motivate the proposed methods, suggesting that some variants of the proposed methods could likely be substantially better. In addition, I agree strongly with R3 that framing this work around 'language structure' is disingenuous, and actively misleads readers about the contribution to the paper.